# Investigating the Impact of Presentation Format on Reading Ability in Posterior Cortical Atrophy: A Case Study

**DOI:** 10.3390/reports8030160

**Published:** 2025-08-31

**Authors:** Jeremy J. Tree, David R. Playfoot

**Affiliations:** Department of Psychology, Faculty of Medicine, Health and Life Sciences, Swansea University, Swansea SA2 8PP, UK; d.r.playfoot@swansea.ac.uk

**Keywords:** posterior cortical atrophy, attentional dyslexia

## Abstract

**Background and Clinical Significance:** Patients with a neurodegenerative condition known as posterior cortical atrophy (PCA) can present with attention impairments across a variety of cognitive contexts, but the consequences of these are little explored in example of single word reading. **Case Presentation:** We present a detailed single-case study of KL, a local resident of South Wales, a patient diagnosed with posterior cortical atrophy (PCA) in 2018, whose reading and letter-naming abilities are selectively disrupted under non-canonical visual presentations. In particular, KL shows significantly impaired accuracy performance when reading words presented in tilted (rotated 90°) format. By contrast, his reading under conventional horizontal (canonical) presentation is nearly flawless. Whilst other presentation formats including, *mixed-case* text (e.g., TaBLe) and vertical (marquee) format led to only mild performance decrements—even though mixed-case formats are generally thought to increase attentional ‘crowding’ effects. **Discussion:** These findings indicate that impairments of word reading can emerge in PCA when visual-attentional demands are sufficiently high, and access to ‘top down’ orthographic information is severely attenuated. Next, we explored a cardinal feature of *attentional dyslexia*, namely the word–letter reading dissociation in which word reading is superior to letter-in-string naming. In KL, a similar dissociative pattern could be provoked by non-canonical formats. That is, conditions that similarly disrupted his word reading led to a pronounced disparity between word and letter-in-string naming performance. Moreover, different orientation formats revealed the availability (or otherwise) of distinct compensatory strategies. KL successfully relied on an oral (letter by letter) spelling strategy when reading vertically presented words or naming letters-in-strings, whereas he had no ability to engage compensatory mental rotation processes for tilted text. Thus, the observed impact of non-canonical presentations was moderated by the success or failure of alternative compensatory strategies. **Conclusions:** Importantly, our results suggest that an attentional ‘dyslexia-like’ profile can be *unmasked* in PCA under sufficiently taxing visual-attentional conditions. This approach may prove useful in clinical assessment, highlighting subtle reading impairments that conventional testing might overlook.

## 1. Introduction and Clinical Significance

Following brain injury, patients can develop striking acquired dyslexias [1,2]. Some lose the ability to read irregularly spelled words, a pattern known as *surface dyslexia* [3,4,5,6]. Others show disproportionate difficulty reading pronounceable nonwords, reflecting *phonological dyslexia* [7,8,9,10]. Still others make severe semantic errors when reading, indicative of *deep dyslexia* [11,12]. In all these cases, the impairment has traditionally been described as a *central dyslexia*, meaning a disruption to the specialised cognitive system for word recognition [13]. However, reading is also a visually and attentionally demanding activity, and thus a reading impairment may sometimes arise from a more general visuospatial or attentional deficit. Such deficits are termed *peripheral dyslexias* because they imply a non-language-specific impairment in visual processing or attention. One well-documented example is *attentional dyslexia*, introduced in [14], which described patients who manifested a striking dissociation. Namely, they were well-able to read words despite being impaired at naming the individual letters within those same words (i.e., letter-in-string naming). According to their account, a breakdown of the attentional system causes patients to “*have difficulty identifying multiple items of the same type in the visual field… [despite being] …perfect at identifying single items. In this respect, a word counts as a single item, but the letters of which it is composed count as multiple items*” [15]. Consistent with this explanation, similar impairments arise whenever multiple stimuli of the same category (letters, numbers, words, or pictures) are presented together, suggesting a category-specific overload in perceptual–semantic processing [16,17]. In other words, the underlying issue is attributed to a failure of selective-attention control mechanisms that normally allow one to identify and relay several visual items in parallel to higher-level recognition [16,18].

If the inability to report individual letters within a word reflects a difficulty in distributing attention across multiple elements of the string (rather than an inability to recognise single letters, per se [18]), then systematically increasing the attentional demands of reading should amplify the impairment. One potential way to increase attentional demand—the focus of our current work—is to present text in *non-canonical orientations*. For example, reading vertical text (marquee-style, top-to-bottom) often necessitates a sequential, letter-by-letter scanning strategy, and reading rotated text may require mental rotation of the visual input. Both of these requirements can strain attentional resources and thus have the potential to reveal underlying deficits in patients with fragile visual attention [19]. Similarly, using mixed-case formatting (alternating uppercase and lowercase letters within a word, as in *TaBLe*) is argued to increase attentional load or processing difficulty [20,21], albeit for somewhat different reasons than rotated or vertical text (discussed below).

The current work explores how presentation format (both in relation to orientation and letter case) impacted reading and letter-in-string naming in a patient with frank attentional impairments. KL presents with posterior cortical atrophy (PCA), a neurodegenerative syndrome (PCA), which is characterised by progressive deterioration of visual and spatial cognitive functions as a consequence of parietal and occipital degeneration. A hallmark of PCA is a decline in visuospatial attention over the course of the disease, even as memory and language are initially spared [20,22]. Indeed, PCA cases have been documented with a variety of acquired dyslexic reading patterns, including *pure alexia* [22,23], *neglect dyslexia* [24], *spatial alexia* [22], and “*apperceptive*” *alexia* [25]. This diversity of reading profiles likely reflects the phenotypic and pathological heterogeneity of PCA; for example, distinct variants have been proposed, involving predominantly dorsal (parietal), ventral (occipitotemporal), or primary visual cortex degeneration [26]. Nevertheless, PCA commonly involves atrophy in occipitoparietal regions crucial for visual attention, spatial processing, and the integration of visual information [27]. As such, it provides a unique context for studying attentional dyslexia [20]. By examining KL’s reading performance under varied presentation formats—horizontal vs. vertical vs. tilted text and same-case vs. mixed-case lettering—we can aim to *unmask* latent attentional deficits that remain hidden during conventional (canonical) assessments [28]. This approach also sheds light on any format-dependent compensatory strategies and the specific cognitive resources that support reading under increased visual load. For instance, patients might adopt oral spelling or mental rotation to overcome format-induced challenges; the success or failure of these strategies can reveal where processing bottlenecks lie. Having outlined our aims, we now review the relevant literature on each of these key issues: (a) attentional effects of non-canonical presentation formats; (b) the impact of case mixing on reading, particularly in attentional dyslexia and related conditions; and (c) a specific discussion of the nature of posterior cortical atrophy and its specific relevance to the types of reading testing we aimed to undertake.

### 1.1. Attentional Effects in Word and Letter-in-String Naming: Non-Canonical Presentation Formats

The current work is inspired by a key study undertaken by the authors of [29], who described a neurodegenerative case with predominant dorsal (parietal) atrophy with consequential attentional deficits. Interestingly, despite largely intact *canonical* word reading (i.e., words presented in a typical format), their case became *severely* impaired in reading when non-canonical presentation formats were employed. That is, against a baseline of intact horizontal reading, the patient’s performance fell to near floor level for both vertical and 80°/90°—rotated word—presentations, with no notable difference between them. In addition, the consequences of manipulating the rotation angle appeared non-linear: for rotations beyond 50° or greater, reading accuracy dropped precipitously (a phenomenon also observed for reaction times in healthy readers; see [30]). This pattern indicates that word reading clearly becomes severely compromised after a certain “critical angle” of rotation. The authors of [29] interpreted their findings as clearly indicating that non-canonical presentations for reading must result in *increased attentional load* that imply “dorsal” (parietal)-lobe reading processes—such processes being disrupted in their case. Put simply, it was argued that their findings suggest that patients with attentional limitations may appear to read accurately in typical clinical (canonical) testing conditions, but when spatial orientation demands increase, latent deficits of attention on reading can be “unmasked”.

Other work with neurologically normal readers has also shown that unusual text orientations can disrupt visual word recognition. For example, [31] explored how tilted and vertical/marquee presentations impacted visual word recognition in healthy adult readers (see Figure 1). Interestingly, the authors found that of these two non-standard formats, vertical text was substantially more disruptive to word recognition than tilted text. The authors interpreted this pattern as evidence that word recognition normally relies on coding letters with respect to a principal horizontal “axis” of the word (i.e., an object-centred reference frame aligned with the word’s usual orientation) rather than purely in viewer-centred coordinates. Consequentially, disruption to this *principal axis* resulted in much higher “attentional load” and thus poorer performance. This is consistent with [32], which reported that the reading speed for vertically arranged text was about 1.8 times slower than for standard horizontal text and roughly 1.3 times slower than for text rotated 90°—again indicating that vertical format incurs the greatest processing cost.

Inspired by these studies, we applied similar presentation-format manipulations to our PCA case study, of KL, to determine if word reading would be disrupted. In addition, we investigated whether an attentional dyslexia pattern—specifically, better performance in whole-word reading than in letter-by-letter reporting—would also emerge in KL when the format was *non-canonical*. Finally, KL’s ability to compensate under some non-standard conditions but not others could offer insights into the cognitive mechanisms supporting residual reading ability. For instance, if KL could adjust to certain rotations but not to vertical text (or vice versa), this would indicate not just the utility of different strategies but the prospect of dissociated patterns of disruption. In addition to manipulating non-canonical presentations on KL, we also aimed to manipulate letter case by presenting mixed-case letter strings. This manipulation has also been claimed to increase attentional load for reading tasks and provided further determination of the impact of such manipulations on KL’s reading and letter-in-string naming.

### 1.2. The Impact of Case Mixing on Reading in Attentional Dyslexia

As mentioned earlier, a robust finding is that mixed-case presentation—such as alternating uppercase and lowercase letters within a word (e.g., *TaBLe*)—impairs reading in attentional dyslexics [33]. This disruption has traditionally been attributed to an interference with holistic visual word-form representation. In other words, alternating case disrupts the familiar overall shape of the word, making it harder to recognise the word as a single unit. The authors of [34] noted that while words in a normal uniform-case format can be processed rapidly in parallel (showing little effect of word length on reading time), introducing distortions such as case mixing, increased letter spacing, or vertical text tends to slow down reading and reintroduce word-length effects—consistent with a shift towards more sequential processing under these conditions. Other researchers have challenged the suggestion of a “holistic” disruption account to case-mixing manipulation. If the disruption were solely due to losing the word’s overall shape, then increasing the spacing between the letters—which also disrupts the holistic word form—should not *improve* reading. Yet studies have shown that adding extra letter spacing can actually facilitate reading under crowding conditions by reducing visual letter interference [35]. As a result, an alternative interpretation has been suggested for the consequences of mixed-case manipulations: lateral masking (crowding) and incorrect letter grouping. In mixed-case words, uppercase and lowercase letters have different sizes and extents, so a tall uppercase letter can spatially “mask” or “crowd” adjacent smaller lowercase letters, impeding their identification [35,36]. Indeed, consistent with this logic, increasing letter spacing (which alleviates crowding) tends to improve letter-in-string identification in mixed-case formats, whereas manipulations that restore a consistent letter shape (such as using a uniform size for all letters) do not fully restore performance if the sequence still alternates case [35].

However, the debate on the degree to which mixed-case manipulations favour crowding or “holistic” (orthographic form) positions remains unclear. For example, the authors of [1] reported that their attentional dyslexic case (GK) was markedly impaired when reading mixed-case words, yet showed significantly better performance when the same words were presented in uniform case blocks. Importantly, this benefit of consistent casing was observed even when letter size differences were eliminated. In other words, making all letters the same height did not improve GK’s mixed-case reading; he only improved when the letters were truly uniform in case. This finding argues against an explanation based purely on lateral masking from large letters onto small ones (since the size was controlled) and instead implicates the disruption of same-case letter “units.” GK’s reading appeared to rely heavily on maintaining consecutive letters in the same case (and thus the same shape); breaking that continuity via case alternation was enough to derail his word recognition and reading. Additionally, GK was unable to read nonwords at all, underscoring his reliance on accessing stored orthographic representations in his ability to read real words. In any case, the debates around the underlying causes of reading disruption in mixed-case presentation aside, it is apparent that such a presentation manipulation can have implications for reading and letter-in-string naming for individuals with attentional dyslexia. We are unaware of any study with such patients that has contrasted this manipulation with other “attentionally demanding” conditions, and thus our study seeks to redress this omission.

### 1.3. The Current Study: Attentional Dyslexia and Posterior Cortical Atrophy

Posterior cortical atrophy (PCA) is a neurodegenerative condition—often considered an atypical variant of Alzheimer’s disease—defined by the gradual degeneration of the occipitoparietal and occipitotemporal regions of the brain. Clinically, PCA presents with progressive deficits in visual perception, visuospatial skills, reading, and other posterior cognitive functions, while early memory and insight are largely preserved [27,37]. Given the parietal-lobe involvement, PCA patients frequently experience reductions in attentional capacity and visual processing efficiency as the disease advances [20]. This makes PCA a particularly informative model for studying how increased attentional demands can affect reading and whether phenomena like attentional dyslexia might emerge in such patients.

In the present study, we examine patient KL, a gentleman with PCA who exhibits relatively preserved language abilities but significant visuospatial and attentional deficits. We assessed KL’s performance in reading words and naming letters under a variety of challenging conditions designed to tax visual attention. Inspired by the seminal work of [29], we employed two key non-canonical presentation formats to determine the consequences for KL’s reading and letter-in-string naming. In addition, we also manipulated the letter case of the text (comparing same-case vs. mixed-case words) across the same tasks. Although his word reading was intact under typical (canonical) clinical assessment conditions, our aim was to determine whether KL’s reading profile would mimic classic attentional dyslexia symptoms under these taxing conditions—that is, showing a disproportionate impairment in letter-by-letter identification relative to whole-word reading, particularly when the normal cues for parallel processing (canonical orientation, consistent letter shape, lexical familiarity) were disrupted. Based on the literature reviewed above, we hypothesised that KL would read words normally in the standard horizontal format but would show emerging deficits with vertical and rotated text, as well as with mixed-case words, reflecting the *unmasking* of latent attentional limitations. However, the degree to which these different manipulations may provoke different degrees of *relative* disruption and, if so, which may be more or less disruptive was unclear. Equivalently, we further expected such challenging formats to induce the cardinal feature of attentional dyslexia (i.e., poor letter-in-string naming), but again, it remained an open question as to whether all manipulations would provoke similar degrees of impairment. In sum, this case study investigates how orientation and visual-format disruptions impact word reading versus letter-in-string naming in PCA, offering insight into both the nature of attentional dyslexia symptoms and potential clinical approaches for revealing subtle attentionally based reading deficits in the neurodegenerative condition known as PCA.

## 2. Case Presentation

Posterior cortical atrophy (PCA) is diagnosed on the basis of *insidious onset and gradual progression* of symptoms, with *early, prominent disturbance of visual or posterior cognitive functions* and relative sparing of memory, language, and executive abilities in the initial stages [37]. Characteristic early diagnostic criteria include cognitive deficits such as visuospatial disorientation, simultanagnosia, object or face recognition problems, constructional and dressing apraxia, and alexia or acalculia. Supportive features are *posterior cortical atrophy on MRI* or hypometabolism on PET, while the exclusion criteria include alternative structural or ophthalmological causes and presentations dominated by memory or language decline. Although PCA is a *syndrome*, in most cases, it reflects an atypical posterior variant of Alzheimer’s disease [37,38].

KL was a 62-year-old right-handed man with 16 years of formal education. He had 4 years of progressive visual difficulty affecting his work as a computer programmer. He could no longer read code because characters would “*move around on the screen*”. He denied problems with language or episodic memory, though he found that his “*thought processes were slower*” and he had difficulty with planning. A SPECT scan (undertaken in May 2018) had shown reduced cerebral perfusion in the parietal and posterior temporal lobes, consistent with PCA; in October 2019, KL underwent a brain MRI that demonstrated predominant occipitoparietal- and/or occipitotemporal-lobe atrophy with widening of the parieto-occipital sulci on visual inspection, consistent with a PCA profile [38]. KL was included in a recent case-series publication exploring mental imagery abilities in PCA [39], in which KL demonstrated severe impairments. A full report of all initial screening in neuropsychological testing (undertaken across several sessions in September–October 2019) is provided in that work, so to avoid unnecessary duplication, we suggest that interested readers refer to that paper. Suffice it to say KL’s neuropsychological testing confirmed specific attentionally based object and visuospatial processing deficits with relative sparing of other cognitive domains, such as language and memory (consistent with a PCA cognitive profile). Given our specific focus on reading in this case, in that screening period, his word reading (and related abilities) was evaluated using tests from the PALPA (Psycholinguistic Assessments of Language Processing in Aphasia, [40]), with word-reading accuracy being excellent on lists varying in regularity (PALPA 35-60/60) and frequency/imageability (PALPA 31-80/80). In addition, nonword reading showed a mild impairment (PALPA 36-21/24); interestingly, emerging nonword-reading problems (phonological dyslexia) have been linked to emerging attentional problems [41,42], which has been interpreted as indication that grapheme–phoneme processes linked to letter scanning may be disrupted. By comparison, the case described in [29] scored 18/20 on word reading and 6/20 on nonword reading. Other reading-related testing with KL included visual word lexical decision (PALPA 25-120/120), written rhyme judgement (PALPA 32-55/60), and testing of letter naming (26/26 upper/lowercase) and letter matching (26/26 upper/lowercase)—all of which established that KL appeared to have no problems. Oral (35/40) and written spelling (35/40) were also reasonably well-preserved. In sum, KL has cardinal features of PCA linked to impairments of attention but no obvious language or word-reading problems at this stage of his disease progression.

### 2.1. Experiment 1—Word Reading Across Canonical and Non-Canonical Presentation Formats

In our first experiment, we sought to investigate KL’s word reading across different non-canonical presentation formats (marquee and rotated), consistent with the work of [29]. We also included a mixed-case manipulation as an additional comparator to our other attentionally demanding conditions.

#### 2.1.1. Materials and Procedure

In this study, we constructed two sets of 80 words: set A varied letter length by using two sets (40 items each) of 3- vs. 5-letter words, and set B similarly varied letter length with 3- vs. 6-letter words. These were carefully controlled for other key variables such as frequency and imageability. In their work, the authors of [29] reported evidence for a letter length effect for vertically presented words (3 letters = 50% accuracy, 6 letters = 0% accuracy); as a consequence, we included letter length manipulation to determine if a similar pattern was observed for KL (note that they did not manipulate length for rotation, which we included). All items were matched for word frequency across the length manipulations.

All items were presented for in-person testing via a PowerPoint presentation (font size = 80, font type = Calibri, all uppercase), with no time limit for each item and the experimenter moving each item on after a response was generated or KL indicated he had no response to give. Presentation formats (see Figure 2) were deployed in a blocked session format over consecutive weekly sessions, with both Set A and Set B being employed each week (8 weeks of testing total). In all cases, we report accuracy across tasks, presentation formats, and length.

#### 2.1.2. Results

In Table 1, we present the performance of KL reading sets of words across all of our key manipulations. For normal (canonical) presentation, reading is clearly intact, regardless of letter length, consistent with the screening data we mentioned earlier. In other words, no obvious attention-based reading problems were observed in a typical clinical presentation setting. Mixed-case presentation was also equally good, suggesting that a “classic” manipulation claiming to increase “crowding” and thus “attentional load” (see introduction) had little impact. Interestingly, the two non-canonical presentations provoked very different patterns of performance. In the case of marquee presentation, mild disruption was observed with some evidence of a length effect (albeit nowhere as profound as that observed in [29]), whereas tilting 90° had a *profound* impact on reading—that is, we appeared to have a *format-specific* “unmasking” of an attentional impact on single-word reading. The fact that we could see a *selective consequence* of format presentation would suggest that impaired performance cannot simply be dismissed as being due to “crowding” or lateral masking.

Interestingly, for the marquee presentation, it was clear that KL started to utilise an explicit strategy of naming letters in strings to assist his accurate reading. That is, he would often spontaneously name the letters presented, which attenuated his error performance (e.g., sees TABLE and says “*T-A-B-L… must be table*”), and this would be consistent with mildly poorer performance with longer letter strings (akin to a pattern observed with letter-by-letter readers but less striking [34]). In contrast, for tilted items, he would often spontaneously report that letter strings looked “*more like squiggles*”, suggesting a qualitatively different subjective experience of items presented in that manner. Interestingly, the tilted format performance did appear to reveal two other curious observations: (a) that counter-clockwise presentation may have been relatively more disrupted and (b) that performance was generally better with *longer* letter strings (something that was not investigated in the work of [29]); namely, for rotated strings, there is a *reverse length* effect. We will return to both these issues in a later experiment that specifically focuses on tilted performance (see below).

This naturally begs the question as to why the case described in [29] showed profound reading impairments with *both* vertical and rotated items and yet our case, KL, was only impaired with the latter. One possibility may simply be that their case had a much higher degree of general attentional “resource” impairment, such that they were always likely to show some problem with *any* non-canonical manipulation of letter strings. But such a general “attentional resource demand” account would naturally imply, for KL at least, that tilting must involve a *higher* degree of such resources. We do, however, favour an alternative explanation—this is because KL very apparently adopted an explicit letter-by-letter spelling strategy for the marquee presentation (which he did spontaneously) to “augment” his performance. That is, KL was readily able to use an *oral spelling strategy*, given his intact abilities in this domain. Interestingly, this would imply that the case in [29] likely had some additional problems with oral spelling and was thus unable to use a similar *strategy* for such a presentation—although there was no detailed assessment of spelling for the case in [29], those appendices do report that she had a “*central agraphia, as evidenced by the numerous errors she made when spelling out words orally, including her own name*”. So it seems reasonable to infer that a letter-by-letter oral strategy was not available to her to compensate. It is conceivable that KL would have lost the ability to adopt the letter-by-letter strategy as his PCA progressed, eventually presenting a similar pattern of impairment to that of their case [29]. The COVID-19 pandemic interrupted our work with KL, so this remains an open question. Meanwhile, we interpret KL’s profound problem with tilted (rotated items) as being consistent with the fact that in order to read off such items, participants typically *mentally rotate* a letter string to a more canonical orientation. Previous work has suggested that lesions to the parietal lobe can disrupt processes of mental rotation [43,44], including using letter stimuli [45]. And our own recent work [39] has demonstrated that KL (and other PCA cases) were all profoundly impaired in mental imagery tasks. Consequently, KL was unable to use this strategy to assist in reading words presented in a tilted fashion, and thus, a severe impairment naturally followed. In general, it seems that different non-canonical presentations likely provoke the use of additional “strategies” for patients and even healthy controls—and any observation of “preserved” reading in such non-canonical contexts will be heavily dependent on their availability for deployment.

### 2.2. Experiment 2—Focusing on the Impact of Tilted Rotational Presentation on Single-Word Reading

Having established that KL appeared disproportionately impaired at reading tilted text, we undertook a second experiment to further investigate the consequences of rotation. The authors of [29] reported that the impact of rotation was non-linear, in that for their case presentation of letter strings at 10°, the 30° and 50° levels of global rotation had minimal impact. In contrast, 80° and 90° rotation performance was severely impaired in reading 50 five-letter words. In our case, we used 120 five-letter words tested over three sessions that varied across three presentation orientations (normal, tilted 50°, or tilted 90°).

#### 2.2.1. Materials and Procedure

Again, as in the previous experiment, all words selected were matched for frequency and imageability. By systematically tilting words (see Figure 3), we were able to determine if, like the work of [29], the consequences of tilting for KL were non-linear such that performance would drop off dramatically at a certain level of rotation.

#### 2.2.2. Results

Our observations are presented in Table 2 and indicate a pattern consistent with the work of [29]—namely, that performance is good at 50° and drops precipitously at 90°. Ref. [29] suggests that the ventral (lexical) pathway preserves some form of orthographic “perceptual invariance” for rotations of up to 50°, and after this “*critical rotation angle*”, there is some deployment of “parietal” dorsal processing, which we would argue implies the utilisation of some form of mental imagery/rotation strategy. Interestingly, testing of neurologically intact adults has similarly suggested that beyond 60° rotations, readers are unable to utilise the typical reading system [30,46]. Other work observing IT neurons in macaques has similarly indicated an object-invariant neural response for object rotations of up to 50° [47]. For both KL and the case described in [29], the implication is that their parietal-lobe atrophy has severely disrupted this mental imagery system—although unfortunately, in the latter case, no explicit testing of mental imagery ability was undertaken. Interestingly, [29] acknowledges the likely deployment of “strategies” when reading 80°/90° tilted words but chose not to favour a simple mental imagery impairment explanation—the authors wrote: “*This complex nonlinear pattern does not fit a simple hypothesis that reading tilted words would always require mental rotation, with a difficulty directly proportional to the rotation angle…angles greater than about 60*° *require letter-by-letter reading (maybe in combination with mental rotation)…* [bold for emphasis by ourselves]”. Moreover, Ref. [30] concluded that “*the case for mental rotation during reading remains unproven.*”, cautioning that “*it is not clear whether the reading of rotated words involves the same sort of mental rotation process envisioned by Shepard and colleagues*.” Given that we have already established that KL has considerable mental imagery problems and is very capable of using other strategies in different non-canonical circumstances, we would suggest that a straightforward mental rotation impairment can account for the pattern observed. In addition, we also systematically manipulated whether items were rotated *clockwise* or *anti-clockwise*. Interestingly, the authors of [29] reported that their case was much worse with clockwise rotation (32% accuracy) than anti-clockwise (68% accuracy), whilst for KL, it appears that the reverse is true. However, we would suggest caution in interpreting the findings of [29] in this respect, since they only tested 22 items across both cases (despite reporting using a 50-item testing set).

### 2.3. Experiment 3—Naming of Letters in Strings in Non-Canonical Presentation Formats

Having established the relative impact of different forms of non-canonical and mixed-case presentation on word reading, in a third experiment, we were interested in exploring a key feature of attentional dyslexia patients—that is, that they have greater problems reporting the constituent letters of words (and other strings) despite being well-able to read the same words (see introduction). This was not investigated in the work of [29] (or any PCA patients thus far), so this constitutes an entirely novel investigation.

#### 2.3.1. Materials and Procedure

In this case, we selected the 80-word items from set A used in Experiment 1, which comprised 3- and 6-letter items (N = 40 of each). We also largely employed identical presentation formats (see Figure 2 for examples). The key difference was in the instructions—namely, KL was required to “*name each letter in the string out loud*” rather than name the word. In this case, we assumed the task was naturally more attentionally demanding, but it remains an open question as to whether the same relative impacts of different presentation formats would mirror that observed with single-word reading. As an additional element, we also investigated KL’s ability to name number strings of the same lengths as the words (3 vs. 6) under similar presentation manipulations. In this case, KL was instructed to “*name each number in the string out loud*”. In all instances, we reported absolute accuracy: that is, responses where *all constituents* in the string were successfully named.

#### 2.3.2. Results

The observed accuracy levels are presented in Table 3, and it is clear that for words, letter-in-string naming is excellent under normal (canonical) circumstances, whilst for number performance, it is slightly worse for longer digit sequences. There is a mild drop in performance that is equivalent for both vertical and mixed-case presentations, with an identical length effect for words (+12% for 3-letter strings). However, rotated letter strings clearly demonstrated a *profound problem* for KL, which was mirrored for number strings (to a larger extent). Interestingly, it is apparent that letter-in-string naming rotation provoked a typical length effect, whereas there was a *reverse length* effect in reading of the same items (see above). Importantly, the letter and number performance for vertical strings is equivalent, which eliminates the possibility that KL was also naturally boosting his performance via a spelling strategy (as with reading) in this case too.

In any case, reflecting on the summary we present in Table 4, we would argue that the cardinal feature of attentional dyslexia (namely that word reading > letter-in-string naming) is clearly modulated by relative attentional resource demand. Given the lack of such impairments for canonical format presentation, we would argue that this arises because of successful access to orthographic long-term memory information via preserved ventral-route processing. Such “top-down” information also ameliorates any emerging problems that may also arise from a mixed-case format. However, in the case of our two critical non-canonical presentation formats, the typical access to such information was eliminated, and some form of compensatory strategy was required. In the case of KL, in one instance (vertical), a strategy was available for reading but was not sufficient to considerably “boost” letter-in-string naming (and thus an attentional dyslexia profile emerged). In the other instance (rotated), no such strategy was available, and thus performance in both reading and letter-in-string naming was severely impaired (although even here, evidence for an attentional dyslexic-“like” profile was observed). In sum, the use of non-canonical presentations can appear to *unmask* an attentional dyslexia profile in PCA cases such as KL.

## 3. Discussion

### 3.1. Format-Specific Effects and Strategy Use

Under standard canonical text presentation, KL’s reading appeared intact, and similarly, the letter-in-string naming was also normal. However, this changed when we utilised non-canonical presentation formats, specifically tilted presentation. Importantly, we argue that despite quite generally good accuracy for marquee (vertical) presentation, this was largely due to KL’s ability to compensate by adopting an oral letter-by-letter spelling strategy to augment performance. This sequential, phonological strategy allowed him to successfully parse vertical letter strings, effectively performing like a pure alexic (letter-by-letter) reader under this specific format circumstance. This contrasted with the tilted presentation, where no such alternative strategies were available, since we suggest that mental rotation or spatial transformation for recognition is severely impaired in KL [39]. Additionally, we have confirmed the striking rotational “break” that occurred above 60 degrees of tilt, where performance dropped precipitously. The fact that KL did so well at lower levels of tilt despite severe mental imagery impairments suggests that orthographic access remains at such levels. We argue that our findings demonstrate that non-canonical formats can effectively *unmask* apparent attentional problems in the sphere of reading, which (if possible) can be compensated for by deploying specific cognitive strategies, if available: namely, linear scanning and phonological recoding can assist in reading vertical text, and mental imagery is likely to be deployed for 60° + rotated text (consistent with other work discussed earlier). With respect to the only other neuropsychological study of similar manipulations [29], we would argue that both such compensatory strategies were unavailable for their case, and hence the strikingly impaired profile reported was revealed in both vertical and rotational formats. In sum, our findings with KL illustrate how non-canonical formats increase the attentional demands of reading and how compensatory strategies—when available—can offset such demands.

KL’s pattern of performance underscores the importance of “top-down” (ventral) lexical access in reading and how it can be disrupted by unusual orientation formats. Under normal circumstances, his preserved ability to read familiar words suggests that he can efficiently access whole-word orthographic representations from long-term memory. We propose that in typical reading, orientation is an essential “attentional gate” that allows KL (and others like him) to access this ventral route to retrieve stored word-form representations [48]. When words are presented in their customary left-to-right orientation, KL can take advantage of this *top-down support* via ventral neural projections, resulting in fluent reading with minimal errors. However, non-standard orientations (vertical or tilted) appear to block or degrade access to whole-word forms, forcing KL to rely on slower, piecemeal strategies that are subsequently heavily attentionally demanding and linked to ventral neural projections (see also [29]). In vertical format, because letters remain upright, this allows KL to resort to a letter-by-letter decoding approach (analogous to that seen in pure alexia) to make *lexical sense* of the string. The authors of [49] noted that patients with acquired dyslexia will use *any available* means to construct meaning from letters, and KL’s behaviour exemplifies this principle—he engaged whatever strategy was necessary to recognise and read words when the usual route failed.

Notably, changing the case format (alternating upper- and lowercase letters in strings) had relatively little impact on KL’s word reading, in stark contrast to the orientation (tilting) manipulation. This finding suggests that classical visual–attentional factors like crowding or lateral masking were not the primary drivers of his deficits. If crowding were a major issue, one might expect mixed-case text (which can disrupt word shape and letter processing) to have impaired reading as well. Instead, KL’s preserved performance with case-altered words implies that his reading system could tolerate changes in letter shape or size as long as the overall orientation was within canonical parameters. In other words, the critical factor for KL was whether the global orthographic layout was familiar and did not simply reflect *any* visual distortion (such as mixed case). This aligns with the idea that his reading difficulty under non-canonical formats stems from a failure to engage “top-down” (ventral) orthographic knowledge when the stimulus presentation is too atypical. Consistent with this interpretation, prior studies of PCA-related alexia have noted that patients rely heavily on stored word (orthographic) knowledge; Ref. [25] reported that PCA patients read real words more easily than pseudowords and often produced visual–lexical errors, reflecting a dependence on lexical memory. KL appears to capitalise on such “top-down” (ventral) support in canonical reading, even under mixed-case conditions, and when that support is disrupted by orientation changes, his performance declines precipitously. Crucially, the minimal effect of case alternation in KL indicates that his deficit is not a generalised attentional problem for all visual disruptions but a specific inability to access whole-word forms when text orientation deviates beyond a tolerable range. This pattern reinforces that orthographic recognition is orientation-sensitive: standard case changes do not close the *attentional gate* to lexical access, but rotating the text does (see [29]).

Overall, it is apparent that KL differs from both the [29] case, who displayed a profound breakdown in reading across all non-standard presentations, and other cases discussed earlier with classical attentional dyslexia. That is, compared with prior cases, KL demonstrated a uniquely format-sensitive dyslexia underpinned by the ability to harness alternative routes in some situations and a failure to do so in others. This comparative perspective emphasises that attentional dyslexia is not a monolithic syndrome; even within PCA or Balint’s-syndrome patients, there can be significant variability. KL’s profile enriches the literature by showing that the presence of attentional “dyslexia-like” symptoms can depend on task and format demands and that individual patients may have spare cognitive resources to exploit under certain conditions. Such distinctions underscore the importance of detailed, format-varied assessments: without testing beyond the usual horizontal text, KL’s problems under more attentionally demanding conditions might have gone undetected, whereas prior cases would have been identified with even, routine testing.

### 3.2. Theoretical Contributions and Clinical Implications

This single-case study yielded several important theoretical insights and clinical implications. First, it provides clear evidence that attentional “dyslexia-like” symptoms can be *unmasked* by employing non-canonical presentation formats, replicating aspects of the seminal work of [29]. In KL’s case, the key hallmark symptom of attentional dyslexia—disproportionately poor letter-by-letter or letter-in-string identification compared to word reading—only manifested under non-standard orientations (particularly when text was rotated). This finding supports the notion that attentional constraints may remain dormant or compensated until a task demands exceed a certain threshold, at which point the deficit becomes observable. The results thereby extend classical accounts of attentional dyslexia by demonstrating a *latent variant* of this condition in PCA—moreover, it is important to stress that such different presentation formats will also provoke the deployment of different reading compensatory strategies. In other words, the observation of accurate (or otherwise) reading in patients will always involve the complex interplay of the availability of access to orthographic long-term memory information via the ventral stream on one hand and the availability of dorsal-based compensatory strategies on the other. In KL’s case, the evidence suggests that different strategies can be dissociated in impairment, such that mental rotation can be severely impaired and yet letter-by-letter oral spelling can remain intact. We had intended to track the course of neurodegenerative disease progression for KL over time to determine if the latter “intact” strategy would also decline and his presentation would match that of the case described by [29], but sadly, COVID-19 prevented this. In any case, we would argue that our findings align with a dynamic view of reading in which multiple pathways or mechanisms can support performance, and their contributions vary with context. Moreover, further future studies may well investigate similar manipulations across PCA patient populations and other attentional dyslexia populations, such as Balint’s syndrome [41] and developmental dyslexia [50,51].

Secondly, our findings lend support to cognitive models emphasising the role of top–down, ventrally based, lexical activation gated by visual format. KL’s preserved reading of words in normal orientation, contrasted with his substantial decline under rotated conditions, suggests that access to stored word-form representations is highly sensitive to visual-format disruptions. This idea is in line with earlier models (such as the various iterations of the Dual-Route cascaded model) [49,52] that posit that the recognition of words benefits from top–down influences, but only when stimuli are presented in an expected configuration. When orientation is altered beyond a critical point, as in KL’s case, it appears to interrupt the normal feed-forward activation of the word-form system via the ventral stream, forcing reliance on slower, bottom–up assembly of the word from its letters. Our observations also speak to a theoretical debate about rotated text: [30] questioned whether reading rotated words truly involves mental rotation (see also [48]). The stark drop in KL’s performance for tilted words—in conjunction with his known deficit in mental imagery—suggests that at least beyond moderate rotations, successful reading *does* depend on spatial transformation processes. In other words, KL’s inability to read text rotated by 90° indicates that mental rotation (or an equivalent visuospatial correction mechanism) is a crucial component for recognising highly rotated words. This case thereby bridges a gap in understanding how spatial orientation interacts with the reading system, providing empirical support that significant rotations invoke processes outside the routine word recognition pathway.

In terms of clinical implications, the case of KL highlights the value of incorporating non-standard presentation-format reading tests in neuropsychological assessments when attempting to establish attentional impairments in patients (like those with PCA). Standard reading tests in typical orientation might not reveal milder or compensable attentional deficits; however, as KL demonstrates, challenging the patient with vertical text, tilted text, or other unusual formats can expose underlying vulnerabilities in attentional control and visual processing and the availability of particular strategies. This has clear diagnostic implications. Even patients who perform well in standard reading tests may struggle in real-world contexts that emerge under less-than-ideal visual conditions for reading. Identifying such latent deficits is crucial for comprehensive patient care. It also informs rehabilitation strategies. For instance, knowing that KL can read via a letter-by-letter strategy when needed suggests that therapies could strengthen or train this compensatory route for daily life reading challenges (such as when text must be read at an angle, like labels on shelves or screens). Conversely, recognising that KL struggles with rotated information implies that interventions might also focus on improving spatial visualisation skills or, pragmatically, on adjusting the environment to minimise challenging orientations in critical situations (e.g., ensuring important information is presented in easy-to-read orientations). More broadly, this case underlines that distinct presentation formats can provoke additional specific compensatory “strategies”. Vertical reading engaged phonological/spelling (letter by letter) strategies, whereas rotated reading demanded visuospatial manipulation—a dissociation that could guide personalised intervention (targeting the specifically impaired component in each patient). Finally, our findings encourage further research into *format-specific* reading processes. They demonstrate that even in neurologically normal readers, one might test similar conditions that “suppress” the availability of such strategies to augment reading (for example, adding an articulatory suppression task during vertical reading or a visual distraction during rotated reading) to determine the degree to which typical reading employs these compensatory strategies. In conclusion, the study of KL not only advances the theoretical understanding of reading by illustrating the interplay between attention, spatial processing, and lexical access but also emphasises a practical message: evaluating reading in multiple formats can reveal subtle deficits and guide more effective management of neuropsychological reading disorders [50].

## Figures and Tables

**Figure 1 reports-08-00160-f001:**
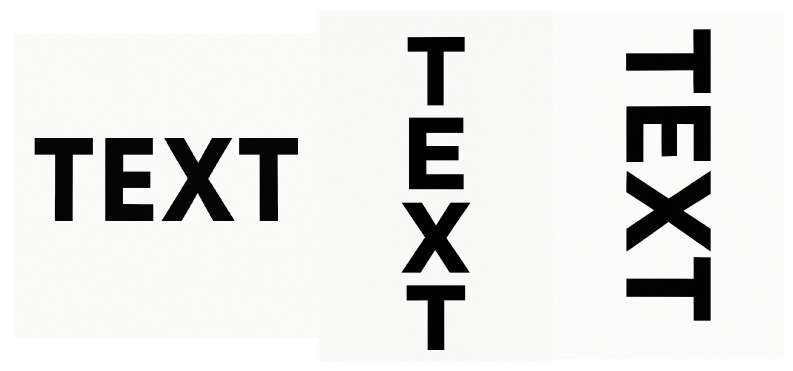
Examples of canonical, vertical, and rotated/tilted text.

**Figure 2 reports-08-00160-f002:**
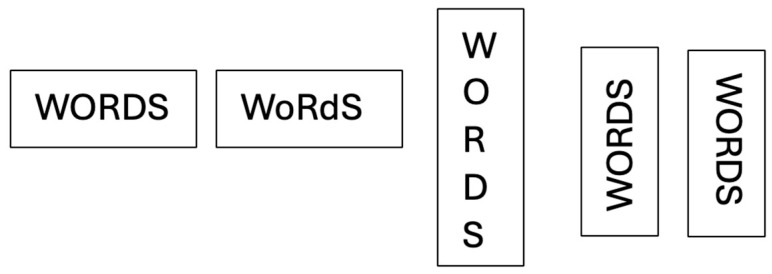
Presentation formats for Experiment 1: canonical, mixed-case, marquee, and tilted.

**Figure 3 reports-08-00160-f003:**
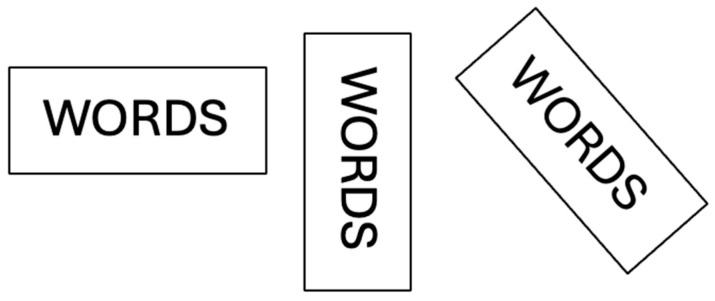
Three presentation formats for five-letter words in Experiment 2.

**Table 1 reports-08-00160-t001:** Reading accuracy across all stimulus sets and presentation formats.

Condition	Accuracy (Set A)	Accuracy (Set B)
Canonical—3 letters	100	97.5
Canonical—6 letters/5 letters	95.0	97.5
Marquee—3 letters	92.5	90.0
Marquee—6 letters/5 letters	87.5	80.0
Mixed case	97.5	97.5
Tilted—3 letters, clockwise	**27.5**	—
Tilted—6 letters, clockwise	62.5	—
Tilted—3 letters, counter	—	47.5
Tilted—6 letters, counter	—	62.5

Note. Accuracy values are percentages. Bold values indicate severely impaired accuracy scores.

**Table 2 reports-08-00160-t002:** Reading accuracy across all three levels of presentation format.

Presentation Format	Clockwise	Anti-Clockwise	Total Accuracy
Canonical	96	99	98
Half tilt (50°)	93	93	93
Full tilt (90°)	65	44	55

Note. Values are percentages.

**Table 3 reports-08-00160-t003:** Letter- and number-in-string naming across stimulus types and presentation format conditions.

Stimulus Type	Condition	3 Items	6 Items	Total Accuracy
Letters	Normal	95	93	94
	Mixed Case	90	78	84
	Vertical	90	78	84
	Rotated	**40**	**20**	**30**
Numbers	Normal	100	70	85
	Vertical	97	87	92
	Rotated	**13**	**3**	**8**

Note. Values are percentages. Bold values indicate severely impaired accuracy scores.

**Table 4 reports-08-00160-t004:** Overall summary across Experiments 1 and 3.

Format	Letter Accuracy (%)	Reading Accuracy (%)	Difference (Reading—Letters)
Canonical	100	97.5	−4
Mixed Case	95.0	97.5	−15
Vertical	92.5	90.0	−23
Rotated	87.5	80.0	−15

## Data Availability

The data and analysis code presented in this study are available on request to the corresponding author.

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
