# Peer review of "Investigating the Impact of Presentation Format on Reading Ability in Posterior Cortical Atrophy: A Case Study"

_reports, 2025, doi:10.3390/reports8030160_

Round 1

Reviewer 1 Report

Comments and Suggestions for Authors

This is a very detailed case report that might indeed help in differential diagnosis of learning disabilities. There are, however, a few corrections and comments. First, do not use references in the abstract because, consequently, your first in-text citation begins with 3. I also think that more information is needed in the abstract about the patient: what was the reason for the consultation? Where did it take place? How long did it last? There are several guidelines for reporting essential data for case studies that authors might benefit from. Moreover, in the first section of the introduction, it would be pedagogical to include images of the ways in which letters and words can be presented in such cases. This could also assist in the right flow of information. This is also valid for the experiments described.

Section 2 needs to include the dates of all the exams and tests. There is a mixture of discussion under case description. I was wondering why not move this into section 3? Also, when results of each experiment are presented, some begin without an introductory sentence. Tables must be revised in terms of formatting too. Finally, the models cited in section 3.2 could be named and explained briefly, and some of the content under this subheading has been mentioned already. In summary, there is room for improvement of the text aiming at its readability for a larger audience, making it more pedagogical and attractive. The date of approval of the IRB is missing.

Author Response

Reviewer 1 – you comments are put as a numerical list with our responses in italics below - we want to sincerely thank you for your time.

Abstract

  1. Remove references – Do not use citations in the abstract, as it causes the first in-text reference to begin with number 3.

We have since removed those citations on your request – apologies for this error.

2. Add more patient details – Include reason for consultation, location, and duration.

This is a somewhat difficult request to make as we did not want to give too much information (as I am sure the reviewer appreciates) that may make it such that others may be able to determine whom this patient is, given their rare neurodegenerative condition. Nonetheless, we concede the point and have added a few elements reiterating some details covered in the case description section – see highlighted yellow in abstract.

Introduction

  1. Add illustrative images – Include images of how letters and words can be presented in such cases to improve clarity and information flow.

This is a good idea and was something that was suggested by reviewer 1 – we have done this and added to the main introduction (see Figure 1).

2. Apply same principle to experiments – For the experiments described, also include such images for better comprehension.

We thank the reviewer for this helpful advice we have now also included two additional figures in the text (see Figure 2 and 3) that provide more information about the format of the stimuli to aid with grasping the format conditions. Thank you for this advice.

Case Description (Section 2)

  1. Add dates for all exams/tests – Ensure that the timing of each assessment is reported.

Again we apologise for that omission as it is a bit rushed, and although all the information is provided in detail in our previous publication (cited) which meant we wanted to avoid unnecessary repetition and redundancy in the scientific record of note, we should have provided some time information for several key elements in the case description (such as the brain imaging) – these are now all included and highlighted in yellow. Thank you for this advice.

Results & Tables

1. Introduce each experiment – Begin each experiment’s results section with an introductory sentence.

We apologise for that oversight and have such sentence – highlighted in yellow – for each of the experimental introductory elements. Again, thank you for that advice.

2. Revise table formatting – Improve formatting for clarity and consistency.

Again, we apologise and acknowledge that our tables could do with some work – so we have totally reformatted all of them (Tables 1-4) on your advice and hope these do a better job for clarity.

Section 3.2 (Models)

1. Name and explain models – Clearly identify the models cited.

We apologise and name the example key model the DRC model of Coltheart and colleagues which is the dominant cognitive model for reading in neuropsychology (see yellow highlighted text) – again apologies for that omission.

Ethics

1. Add IRB approval date – Include the date of ethics committee approval.

We have already provided this to the editor on earlier request – thanks for bringing this to our attention.

Reviewer 2 Report

Comments and Suggestions for Authors

This is a well-written and conceptually rich single-case study of a patient (KL) with Posterior Cortical Atrophy (PCA), which investigates the impact of non-canonical visual presentation formats (vertical, tilted, and mixed-case) on word reading and letter-in-string naming. The manuscript explores how such manipulations may unmask latent attentional dyslexia-like profiles, offering new insights into format-specific cognitive processing and compensatory strategies in neurodegeneration.

The study is methodologically robust, theoretically informed, and makes a clear contribution to clinical neuropsychology, cognitive models of reading, and the understanding of visual attention deficits in PCA.

Comments:  

  1. The terms "canonical", "non-canonical", and "rotated" are used appropriately, but consider a brief visual figure or diagram early in the paper to illustrate these formats (especially for non-specialist readers)
  2. The comparison with Vinckier et al. (2006) is insightful. However, the authors might also benefit from briefly mentioning how KL’s profile might differ from developmental dyslexia or Balint's syndrome, to situate this case in a broader clinical spectrum.

Author Response

Reviewer 2- thank you for your kind positive evaluation of our work and your helpful feedback - below are your comments with our responses in italics beneath them.

The terms "canonical", "non-canonical", and "rotated" are used appropriately, but consider a brief visual figure or diagram early in the paper to illustrate these formats (especially for non-specialist readers).

This is an excellent idea – we have since created an illustrative Figure and inserted it in the introduction section (see Figure 1) to provide guidance as you have suggested. Similar figures (see 2 and 3) have been made for experiment sections on the advice of Reviewer 1.

The comparison with Vinckier et al. (2006) is insightful. However, the authors might also benefit from briefly mentioning how KL’s profile might differ from developmental dyslexia or Balint's syndrome, to situate this case in a broader clinical spectrum.

This is an interesting comment – as neuropsychologists we assume the same underlying functional cognitive model of reading underpins all readers and can be relevant to acquired dyslexias in all contexts. As a consequence, we would assume symptoms of attentional dyslexia in either developmental dyslexia or Balint’s syndrome should be equivalent assuming the same functional cause – we are however not aware of any such study done with individuals with attentional dyslexia across populations in such a context. We do however add a sentence (highlighted in yellow) in the general discussion suggesting this interesting idea and hope you or someone else might explore our similar investigations in this kind of context!

Round 2

Reviewer 1 Report

Comments and Suggestions for Authors

Congratulations for such a speedy, yet robust review. You covered all the issues I had in mind, and your paper reads much more nicely now.

Author Response

Dear Reviewer,

We apologize for the inconvenience caused by the technical issue, which prevented the authors from providing the answer on the website. On behalf of the authors, we would like to sincerely thank you for your time and support in reviewing this manuscript.

Thank you once again.

Sincerely,
Uraiwun Phuangjit
Assistant Editor